# Are 150 km of open sea enough? Gene flow and population differentiation in a bat-pollinated columnar cactus

Sebastián Arenas[1,2], Alberto Búrquez🄳[2]*, Enriquena Bustamante🄳[2], Enrique Scheinvar[3], Luis E. Eguiarte[3]

**1** DIADE, Université de Montpellier, IRD, Montpellier, France, **2** Departamento de Ecología de la Biodiversidad, Instituto de Ecología, Universidad Nacional Autónoma de México, Hermosillo, Sonora, México, **3** Departamento de Ecología Evolutiva, Instituto de Ecología, Universidad Nacional Autónoma de México, Ciudad de México, México

\* montijo@unam.mx

**Data Availability Statement:** All relevant data are within the paper and its Supporting information files. Plastid sequences can be accessed through GenBank accessions KU59180-KU59339 for rpl32-

## Abstract

Genetic differentiations and phylogeographical patterns are controlled by the interplay between spatial isolation and gene flow. To assess the extent of gene flow across an oceanic barrier, we explored the effect of the separation of the peninsula of Baja California on the evolution of mainland and peninsular populations of the long-lived columnar cactus *Stenocereus thurberi*. We analyzed twelve populations throughout the OPC distribution range to assess genetic diversity and structure using chloroplast DNA sequences. Genetic diversity was higher ($H_d = 0.81$), and genetic structure was lower ($G_{ST} = 0.143$) in mainland populations vs peninsular populations ($H_d = 0.71$, $G_{ST} = 0.358$ respectively). Genetic diversity was negatively associated with elevation but positively with rainfall. Two mainland and one peninsular ancestral haplotypes were reconstructed. Peninsular populations were as isolated among them as with mainland populations. Peninsular haplotypes formed a group with one mainland coastal population, and populations across the gulf shared common haplotypes giving support to regular gene flow across the Gulf. Gene flow is likely mediated by bats, the main pollinators and seed dispersers. Niche modeling suggests that during the Last Glacial Maximum (c. 130 ka), OPC populations shrank to southern locations. Currently, *Stenocereus thurberi* populations are expanding, and the species is under population divergence despite ongoing gene flow. Ancestral populations are located on the mainland and although vicariant peninsular populations cannot be ruled out, they are likely the result of gene flow across the seemingly formidable barrier of the Gulf of California. Still, unique haplotypes occur in the peninsula and the mainland, and peninsular populations are more structured than those on the mainland.

## Introduction

The present distribution of species is the consequence of phylogeographic processes, ecological interactions, and events of dispersal and/or vicariance. The geographic range of a species might comprise separate populations leading to allopatric speciation [1, 2]. In addition, there may be founder effects when small number of colonizers result in reduced population genetic

trnL, KU59340-KU59485 for trnL-trnF and KU31391-KU1554 for PetB intron D4 (NCBI, https://www.ncbi.nlm.nih.gov/genbank/).

**Funding:** SA thanks Posgrado en Ciencias Biológicas, Universidad Nacional Autónoma de México, and CONACYT for the M.Sc. studentship (480152). This work was also supported by a grant from Dirección General de Asuntos del Personal Académico, UNAM (DGAPA-PAPIIT project: IN213814) to AB.

**Competing interests:** The authors have declared that no competing interests exist.

variability [3]. Major forces behind the actual shaping of the geographic distribution of species include the movement of land masses through the action of tectonics, dispersal distance, changes in climate across the species range, and biotic interactions [4]. Events such as the glaciations confined species to refugia where they persisted during glacial maxima [5–7], and extant populations in these refugia functioned as reservoirs for recolonization once conditions were suitable for range expansion. Tectonics, climatic shifts, and repeated cycles of glaciation have been proposed as important components in the formation of vicariant populations [8] that might produce speciation or fuse divergent populations as barriers relaxed [9].

As noted by Cavin [10], vicariance is the most straightforward hypothesis to explain disjointed species distributions. For example, climate and tectonics closely match speciation in Cichlid fish speciation and phylogeny [11]. In linear stream systems, using flow and habitat characteristics Cañedo-Argüelles et al. [12] described three kinds of dispersers, these most affected by local factors, intermediate dispersers responding to landscape factors, and strong dispersers with no major pattern at the regional-scale. The former is the most likely to have disjoined populations. High-elevation sky islands have distinct, closely related species that became separated from a single taxon during glacial cycles [13]. For insular systems, the explanation of older taxa in newer islands led to the proposal of vicariant metapopulations as an alternative between long-distance biogeographic dispersal (geodispersal) and ecological short-distance species movements [14, 15].

The evolutionary mechanisms of vicariance and colonization through dispersal, shape population´s genetic structure and led to divergence [1, 2]. However, separating the mechanisms controlling allopatric differentiation is difficult, and represents a major challenge in biogeographical research. Gene flow [16, 17] acts as a major unifying force holding together a common gene pool, and the main forces behind population differentiation are mutation, genetic drift, and natural selection. The latter, drive adaptation, particularly when species have a wide genetic basis, wide distribution range, and are exposed to extensive environmental geographic variability [18]. In any case, for differentiation to occur, reduced gene flow is important.

Because of its length (about 1200 km) and narrowness (40–180 km), the peninsula of Baja California, Mexico is in an almost insular environment [19–21]. Baja California isolation is the result of tectonic processes that separated the peninsula from the mainland by gradually creating new ocean floor during the last 12 million years (late Miocene) [22]. The movements of the tectonic plates of the Pacific and North America caused the gradual emergence of the Gulf of California c. 4–5 million years ago as well as the creation of the Rivera plate through what is known as "strike-slip faults" [23, 24]. These geological episodes led to remarkable ecological changes to the biota of the peninsula and to evolutionary processes of allopatric speciation [19–21, 25].

Although the Gulf of California is not very wide (about 200 km at its widest point and much less among islands of the midriff region), it is recognized as a geographical barrier for many terrestrial species of the Mexican Pacific coast [7, 26]. Plant and animal species derived from ancient populations living before and during the opening of the Gulf probably evolved in geographic isolation under new ecological, geological, and oceanographic determinants. These factors led to a high percentage of endemism in the peninsula and the Gulf islands [20, 27]. Also, Wilder et al. [28] showed that although small islands have unique community assemblages, larger islands are strongly related to the continental sources.

As a result of the peninsular isolation many species show morphological, genetic, or functional differences with mainland species [27]. Within the peninsula, some species have been further isolated by the complex topography of its mountain ranges, the active tectonics, and sea level changes that at times turned what is now the peninsula into archipelagos and

biological islands [19–21, 29]. Evolutionary processes including vicariant differentiation as well as recent events of dispersal have played a key role in population differentiation [30, 31]. A striking example of population genetic differentiation across the Gulf of California, are boojum trees (*Fouquieria columnaris*) showing two distinct clades: a homogeneous peninsular clade and a highly diverse mainland clade [32].

Columnar cacti are prominent elements of the warm drylands of North America. Some species, like the saguaro cactus (*Carnegiea gigantea* (Engelm.) Britton & Rose), or the sahuira (*Stenocereus montanus* (Britton & Rose) Buxb.) are only present in the mainland, either disappeared in the peninsula or never dispersed across the Gulf of California [33]. Other species are only found in the peninsula such as the cochal (*Myrtillocactus cochal* (Orcutt) Britton & Rose), and some narrow endemics provide evidence of speciation by geographic isolation in peninsular conditions. The later include the little studied *Stenocereus eruca* (Brandegee) A.C. Gibson & K.E. Horak, *Lophocereus gatesii* M.E. Jones, and *Stenocereus littoralis* (K. Brandegee) L.W. Lenz. Still other columnar cactus species such as the cardón sahueso and the pitaya agria (*Pachycereus pringlei*, (S. Watson) Britton & Rose, *Stenocereus gummosus* (Engelm.) A.C. Gibson & K.E. Horak, respectively) are found in the peninsula and along a narrow strip of the coastal mainland. The senita (*Lophocereus schottii* (Engelm.) Britton & Rose) and the organ pipe cactus (OPC, *Stenocereus thurberi* (Engelm.) Buxb.) are the most widely distributed species in the mainland and the peninsula. The OPC differs from the senita in being pollinated and dispersed by long-distance flying vertebrates, in particular the nectar feeding bat *Leptonycteris yerbabuenae* and several species of perching birds [33, 34]. This feature makes the OPC a suitable model to test the relative contributions of vicariance and geodispersal in its evolution, and in the evolution of columnar cactus-bat mutualisms.

The variation, structure, and gene flow of mainland populations of *S. thurberi* were studied using nuclear genetic markers by Bustamante et al. [35]. They found moderate genetic structure because of the strong gene flow among the populations. The present study focused on elucidating the relative contributions of vicariance and geodispersal in the biogeographic context of the OPC. To do so, we analyzed the phylogeographic pattern, the genetic diversity and structure, and the historical and ecological relationships of twelve populations, comparing the mainland and Baja Californian populations.

## Materials and methods

### Study plant

*Stenocereus thurberi*, also known as organ pipe cactus (OPC) and pitaya dulce, is a multistemmed 3–8 m tall columnar cactus of the arid regions of northwestern Mexico [26]. Its extant distribution covers most subdivisions of the Sonoran Desert below 1000 m elevation, including northern Sinaloa, most of the state of Sonora in Mexico, scattered localities across southern Arizona in the USA, the lower half of the peninsula of Baja California, and some islands of the Gulf of California (Fig 1). As most large columnar cacti, it is a key element of the desert dynamics providing shelter and food to many organisms [33]. Also, it is of ethnobotanical interest, as its fruits are staple food during the driest and hottest part of the year and their woody remains are used as building materials by native people [33]. It has single, perfect, whitish flowers with nocturnal anthesis [34]. Flowers open shortly after dusk and close during the morning of the next day. *Stenocereus thurberi* usually starts flowering from late April to mid-May, and depending on the geographic location, blooming lasts 8 to 16 weeks [34–36]. During this time, its reproductive success depends primarily on bat pollination and secondarily on hummingbirds, perching birds, and hawkmoths [34]. Later in the season, nectar feeding bats and perching birds disperse their seeds [37].

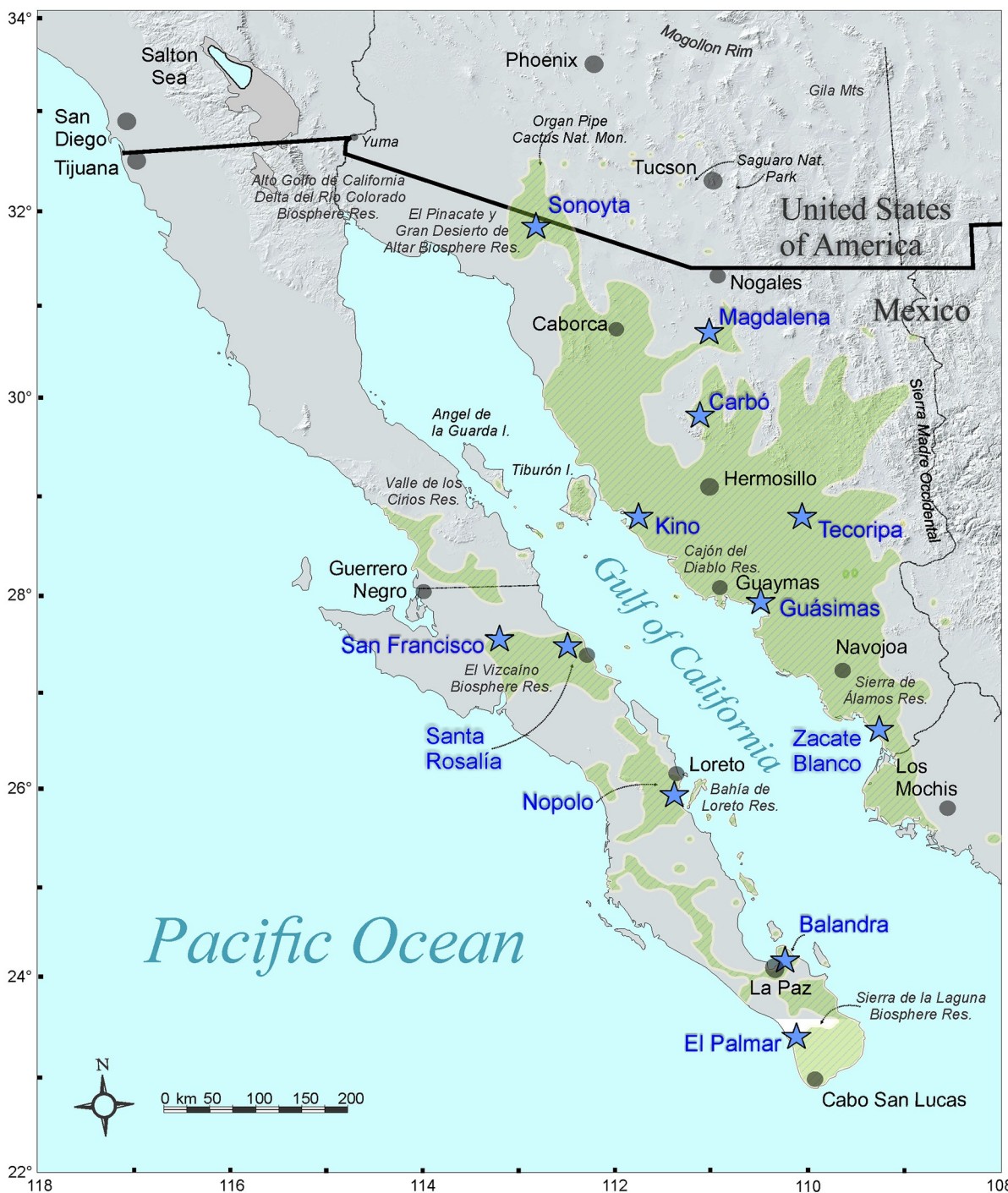

**Fig 1. Distribution of the organ pipe cactus (green shading, modified from Turner et al. [26]).** Populations studied [blue stars, blue font]. Major nature reserves are indicated by italic names. Major cities in grey circles.

## Sampling

Using a clean knife rinsed in alcohol, a thin strip (8x1.5x0.2 cm) of green, heathy tissue was cut from the stem of 20 to 25 adult individuals under permit SEMARNAT 08-017-A. Specimens were arbitrarily selected across a site of at least one ha on each of the 12 sampled populations.

Seven covering the mainland distribution range (mainland group, MAIN from now on), and five populations in the peninsula of Baja California (peninsular group, PEN from now on), ranging from the southernmost population near Los Cabos to the Vizcaino region (Fig 1; S1 Table). Plant material collected in the field from MAIN populations was stored in an ice chest in the field and frozen at the end of the day. Peninsular samples were dry stored in silica gel after field collection. Once in the lab, all samples were stored at -80 ˚C.

## DNA extraction, PCR amplification, and sequence alignment

Total genomic DNA extractions were conducted using a modified cetyltrimethylammonium bromide (CTAB) protocol [for details see 35]. We obtained the total DNA from a set of 15–20 individuals per population. Three non-coding chloroplast fragments (two polymorphic intergenic sequences: rpl32-trnL and trnL-trnF [38] and petB intron D4 gene [39]) were PCR amplified and sequenced from a total of 216 individuals (see S1 Methods for the full PCR procedure).

After removing some nucleotides in the initial and final portion of the sequences (~15 nucleotides), multiple alignments of the sequences were obtained with MUSCLE [40] for a total of 176 samples, and the regions were concatenated with DnaSP v6 [40]. All sequences were deposited in the NCBI GenBank (rpl32-trnL: KU59180-KU59339, trnL-trnF: KU59340-KU59485 and PetB intron D4 gene: KU31391-KU1554).

## Genetic diversity, haplotype phylogeny and network

Diversity indices were calculated for each population and geographic group (i. e., MAIN and PEN groups) as measures of genetic variation for cpDNA. Using DnaSP v6 [41] we estimated the number of segregating sites ($S$), observed number of haplotype ($h$), average nucleotide diversity ($\pi$), and average haplotype diversity [$Hd$; 40]. Significance was evaluated using the coalescent simulation implemented in DnaSP with 10,000 permutations.

Unique haplotypes were selected from a concatenated dataset for phylogenetic analysis with Bayesian inference (BI). We determined the best fit models of nucleotide substitution using jModeltest v0.1.1 [42] under Akaike Information Criterion (AIC). Bayesian analysis was performed with MrBayes v3.2.1 [43] under the following partition: trnL-rpL32 (TN93+G; AIC = 6799.8; lnL = -3223.6); trnL-trnF (GTR+G; AIC = 6302.2; lnL = -3119.1); PetB intron D4 gene (TN93+I; AIC = 2719.4; lnL = -999.7). Chloroplast sequences of *S. gummosus* were downloaded from Arias et al. [44], Hernández-Hernández et al. [45] and Plume et al. [46] and used as an outgroup to root the phylogenetic tree (see S1 Methods for the full Bayesian inference).

We implemented a median-joining [47] method to build an unrooted haplotype network using coalescent simulations in Network v.10.2.0.0 (available on www.fluxus-engineering.com, 2020). The intra-specific relationships between haplotypes were analyzed using the least cost criterion, treating gaps as single evolutionary events and indels as a fifth state of character [48].

## Genetic structure

The observed genetic variation among and within populations and among PEN and MAIN groups was partitioned by a hierarchical analysis of molecular variance (AMOVA) using ARLEQUIN v.3.5.1.2 [49]. Three hierarchical divisions based on the genetic variance were identified: a) within populations, b) among populations within groups, and c) among groups using a non-parametric permutation procedure incorporating 10,000 iterations. Pairwise $F_{ST}$ [50] were calculated using ARLEQUIN v.3.5.1.2. Then, a Mantel test was performed to assess the correlation between the genetic distance $F_{ST} / (1—F_{ST})$ matrix and the geoid distances

(km) matrix. In addition, estimates of $G_{ST}$ structuring were calculated for each of the two population groups and for the whole sample. To determine the presence and hierarchy of barriers to gene flow occurring among populations, we analyzed the Nei's matrix of genetic distances with the Monmonier maximum difference algorithm [51]. BARRIER 2.2 was used to determine hypothetical barriers within the distribution of populations.

To study the genetic structure of *S. thurberi*, we used the model-based Bayesian clustering method of STRUCTURE v2.3.4 [52].

## Coalescent inferences of gene flow and divergence timing

Using an isolation with migration (IM) model implemented in IMa v2.0 [53], we estimated the effective population sizes ($N_e$) of each population group, and the gene flow between the PEN and MAIN populations using the relative rates of population migration (*M*) between groups to capture the population dynamics of the population groups in the early stages of differentiation [54], This model uses a Bayesian coalescent-based Markov chain Monte Carlo (MCMC) method to estimate the posterior probability density of parameters [55]. The model involves several simplifying assumptions, such as neutrality and non-recombination within loci, lack of genetic contribution of unsampled populations, and random mating in ancestral and descendant populations [53]. Preliminary runs were performed to assess the convergence of the MCMC chains on the data stationary distribution and optimize upper bounds on prior distributions (*q* = 30, *t* = 5, *m* = 50; where *q* = population size, *t* = divergence time, and *m* = migration rate) and to optimize heating schemes. Final analyses consisted of three runs of 50 geometrically heated chains with burn-in of $5 \times 10^6$ steps. The heating scheme used a geometric model with parameters $h_a$ = 0.9 and $h_b$ = 0.5. A total of $1 \times 10^6$ genealogies were saved after the three long runs and used to calculate parameter values and likelihood ratio tests of models [56]. Demographic parameters were scaled at generation time and neutral mutation rate. We used a generation time of twenty years. In the absence of a well-calibrated estimate for the *Stenocereus* genus, we applied a generic mean mutation rate (μ) of $7 \times 10^{-9}$ base substitution per site per generation for the cpDNA [57], and the average geometric mutation rate of the markers was used to scale the outputs to demographic units.

We used BEAST v2 [58] to produce a calibrated tree over time sequenced regions of cpDNA. This analysis was performed using the best evolution molecular model based on jModelTest 2.1.4 (see results), to determine the approximate dates of the divergence events between the haplotypes of the population groups of *S. thurberi*, i. e., via vicariance (> 2.5 million years ago) or dispersal (<2.5 Ma) using a relaxed clock with a log-normal model and the Yule (pure-birth) speciation process as the tree prior [59]. To calibrate the tree in years, we used one calibration point estimated by Hernández-Hernández et al. [60]. A posterior distribution of time-calibrated trees was estimated in a $100 \times 10^6$ generation MCMC run, with samples taken every 1000 generations. The resulting trees were summarized using TREEANNOTATOR from the BEAST package of R.

## Ecological Niche Model (ENM)

To model the ecological niche (ENM) and to reconstruct the potential geographic distribution of *S. thurberi* in different historical periods we used Maxent v3.3 [61]. A total of 252 occurrences covering the species distribution range were used. These included the populations of the present study, the geo-referenced populations in Bustamante et al. [35], and the records of the GBIF database (http://www.gbif.org/). To avoid redundancy and overfitting, we removed repeated occurrences with the same geographic coordinates and aimed to attain as homogeneous cover as possible. We used 19 bioclimatic layers with a spatial resolution of 2.5 arc-

minute (WorldClim database v1.4) [62] to describe the general conditions of the area at the different modeled times. Highly correlated variables were excluded (Pearson's $r \geq 0.7$), variables with the larger contribution to model development and likely more biological importance were retained (S2 Table). As a result, climatic conditions were measured as a function of 9 bioclimatic variables. These bioclim variables were used to identify suitable actual climatic areas (1960–1990), possible refuges in the paleodistribution of last glacial maximum (LGM, *c*. 21 ka; climate system model 4, CCSM4), and suitable environments during the last interglacial (LIG, *c*. 120–140 ka; climate system mode NCAR-CCSM). We assumed that climatic preferences did not change over time. Model validation was performed using the default Maxent v3.3. [61] configuration with 10 independent subsample replicas.

To determine the role of environmental variation across the distribution of *S. thurberi* on the geographic patterns of retained genetic diversity, we performed a multiple linear regression using SPSS 20 (SPSS Statistics for Windows, Version 20. Armonk, NY: IBM Corp.). We used the 12 populations as replicates, the averaged genetic diversities (*Hd* and π) calculated in DNAsp as response variables, and the most important variables from the ENM as explanatory variables. Models were tested for the pooled populations and separately for the MAIN and PEN populations.

## Results

### Chloroplast data, haplotype phylogeny and network

The three analyzed chloroplast non-coding regions gave rise to sequences spanning 2338 bp. We found 10 segregating sites with 12 different haplotypes (H1 to H12) for the 12 populations (Table 1). Nucleotide diversity (π) ranged from zero to moderate ($0.9 \times 10^{-3}$), and haplotype diversity (*Hd*) from zero to 0.76. The lowest diversity was found in northern Baja Californian populations. Pooled haplotype diversity and average population diversity in the mainland (0.81 and 0.68, respectively) was higher than peninsular haplotype diversity (0.72 and 0.47, respectively, Table 1).

*Tajima's D* tests showed that most populations are neutral, without indications of selection or population changes. Only two populations (Sonoyta and Nopolo) displayed a slight but significant deviation from neutrality (Table 1). There was no association between haplotype diversity and topographic variables. However, when MAIN and PEN groups were analyzed separately, a significant negative association with elevation was found for the PEN group but not for the MAIN group (S1 Fig). Also, a highly significant relationship between *Hd* and annual rainfall showed that *Hd* rapidly increased with rainfall to reach a plateau at about 400 mm annual rainfall (S1 Fig).

A phylogenetic reconstruction estimated by Bayesian analysis partially resolved the haplotype relationships and revealed two main clades: one comprising the mainland haplotypes H1, H2, H3, H5, H6, H8, and H9, and the other peninsular haplotypes H11 and H12 (Fig 2A). Both clades are well-supported (posterior probabilities of 0.75 and 0.96, respectively). Within the large group, some divergent haplotypes also segregated into a mostly Baja Californian clade (H1, H3, and H8) with high support (0.99), and the remaining haplotypes included mainland and peninsular haplotypes. The remaining haplotypes (H4, H7, and H10) clustered with all the above forming a monophyletic group (posterior probability of 1.00), but their phylogenetic relationships were not resolved (Fig 2A).

Additionally, the median joining network of 12 haplotypes (Fig 2B) revealed partial divergence, with only one ambiguity between the haplotypes H6 and H7, pointing to gene migration among MAIN and PEN populations. The PEN population of Nopolo shares its three haplotypes with MAIN populations, mainly with the two coastal populations across the Gulf of

**Table 1. Population genetics of *Stenocereus thurberi* in the mainland and the peninsula of Baja California, Mexico.** Number of polymorphic sites (S), haplotype diversity (Hd), nucleotide diversity (π), and *Tajima's D* test of population expansion/contraction. Sample size (N), Hd and π ± standard error of the mean in parentheses.

| Population | N | S | Hd (±SE) | | π×10⁻³ (±SE) | | Tajima's D | Haplotypes (N of samples) |
|---|---|---|---|---|---|---|---|---|
| **Mainland group** | | | | | | | | |
| *Carbó* | 13 | 3 | 0.74 | (0.03) | 0.5 | (0.02) | 0.47$^{Ns}$ | H2 (4), H4 (3), H5 (4), H6 (2) |
| *Guásimas* | 14 | 2 | 0.64 | (0.02) | 0.4 | (0.02) | 0.70$^{Ns}$ | H2 (6), H4 (6), H7 (2) |
| *Kino* | 14 | 3 | 0.67 | (0.02) | 0.6 | (0.03) | 1.02$^{Ns}$ | H2 (5), H4 (5), H8 (4) |
| *Magdalena* | 16 | 3 | 0.76 | (0.01) | 0.6 | (0.01) | 1.20$^{Ns}$ | H2 (2), H4 (4), H5 (5), H6 (5) |
| *Sonoyta* | 17 | 1 | 0.56 | (0.01) | 0.3 | (0.01) | 2.08* | H4 (7), H10 (10) |
| *Tecoripa* | 15 | 2 | 0.68 | (0.02) | 0.4 | (0.02) | 1.13$^{Ns}$ | H2 (4), H4 (5), H11(6) |
| *Zacate Blanco* | 15 | 3 | 0.72 | (0.01) | 0.7 | (0.02) | 1.76$^{Ns}$ | H2 (5), H4 (4), H11(1), H12(5) |
| **Total mainland** | **104** | **8** | **0.81** | **(<0.01)** | **0.7** | **(0.01)** | **-0.17$^{Ns}$** | 9 Haplotypes/2 unique H10, H12 |
| **Peninsular group** | | | | | | | | |
| *Balandra* | 14 | 5 | 0.65 | (0.02) | 0.8 | (0.05) | 0.66$^{Ns}$ | H1 (7), H2(2), H3(5) |
| *Nopolo* | 13 | 4 | 0.70 | (0.02) | 0.9 | (0.03) | 2.12* | H6 (2), H7 (7), H8 (4) |
| *El Palmar* | 15 | 5 | 0.59 | (0.02) | 0.7 | (0.04) | 0.92$^{Ns}$ | H1 (6), H8 (4), H9 (5) |
| *San Francisco* | 14 | 4 | 0.39 | (0.03) | 0.4 | (0.05) | -0.85$^{Ns}$ | H3 (1), H4(2), H8 (11) |
| *Santa Rosalía* | 16 | 0 | 0.00 | (<0.01) | 0.0 | - | 0.00$^{Ns}$ | H8 (16) |
| **Total peninsular** | **72** | **6** | **0.72** | **(<0.01)** | **0.9** | **(0.01)** | **1.72$^{Ns}$** | 7 Haplotypes/2 unique H1, H9 |
| **Total** | **176** | **10** | **0.87** | **(<0.01)** | **1.1** | **(<0.01)** | **-** | **12 Haplotypes** |

$^{Ns}$ = not significant (P > 0.05);

*P < 0.05.

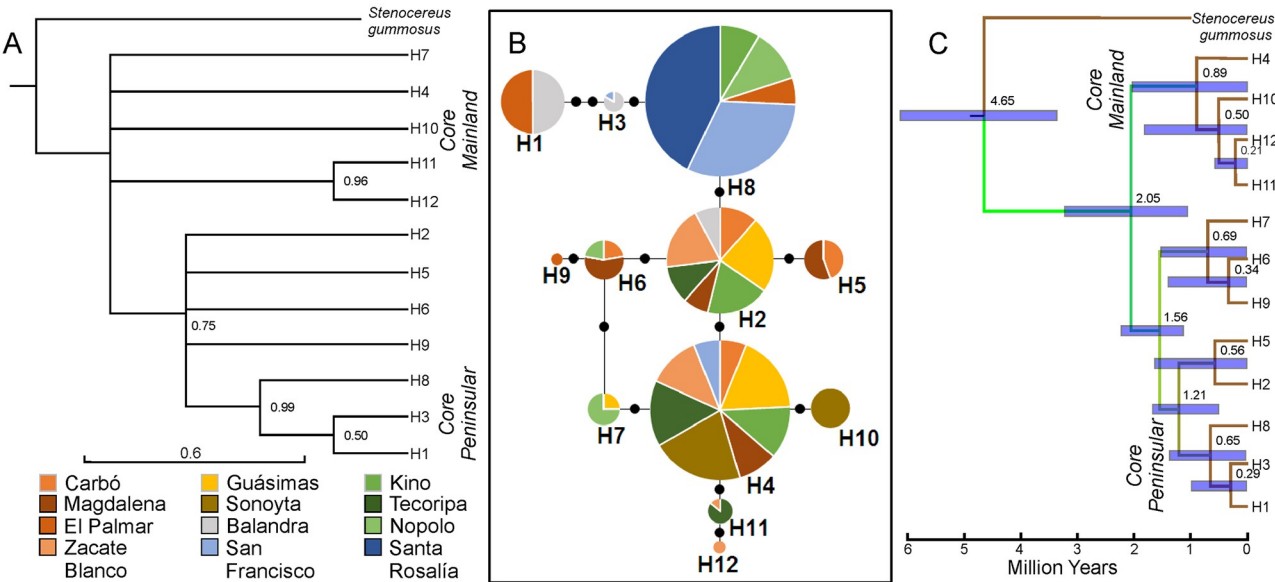

**Fig 2. Bayesian analysis of 12 cpDNA haplotypes of *Stenocereus thurberi* based on three cpDNA regions.** (A). Posterior probabilities are indicated in the branches of the tree. Minimum spanning network of genealogical relationships among haplotypes (B). The size of circles corresponds to the frequency of each haplotype and the small black circles are mutational steps. Phylogenetic tree under a relaxed clock model (C). The estimated times of divergence of nodes are indicated. The blue error bars at the nodes represent the 95% confidence limits.

California (Table 1, Fig 2B, S3 Table). In all cases, the most frequent haplotypes (H2, H4 and H8) were in the center of the network. The least common haplotypes were located at the tips of the network, in the southernmost populations of both the peninsula (H9, El Palmar) and the mainland (H12, Zacate Blanco). Four haplotypes were found only in one population each: two in the mainland at the northern (H10, Sonoyta) and southern (H12, Zacate Blanco) latitudinal distribution of MAIN populations (Table 1), and two in the Cape Region at the southernmost tip of the peninsula (H9 El Palmar and H2 Balandra).

Haplotypes H5, H10, H11, and H12 were only found in the MAIN populations. The northern PEN population of Santa Rosalía had only one haplotype (H8; Table 1), while haplotypes H1 and H9 were only found in the southernmost populations of the Cape Region of the peninsula. Haplotype H7 was the only one that was shared by one coastal MAIN and one PEN population (Fig 2B).

The network showed three main lineages or haplogroups, each derived from a common haplotype: 1) The haplogroup that comprises the H2 and the uncommon haplotypes derived from H2 (H5, H6 and H9), with a wide geographic distribution in the mainland (except in Sonoyta, the northernmost mainland population) and marginally in the peninsula. 2) The H4 haplogroup, including H7, H10, H11 and H12, all of them widely distributed throughout the mainland and almost absent in the peninsula. 3) The H8 haplogroup, also comprising the H1 and H3, found throughout the PEN region and only in the MAIN population of Kino (Fig 2B, S3 Table).

The most probable tree chronogram (Fig 2C) indicated that *S. thurberi* originated *ca*. 2.05 Ma (95% HPD, 1.06–3.23 Ma) during the late Gelasian of the Pleistocene. The largest haplogroup originated about 1.56 Ma (95% HPD, 1.16–2.24 Ma), when the Gulf of California was already well established. It includes three clades, one composed of H6, H7 and H9 haplotypes, diverging roughly 0.69 Ma (95% HPD, 0.009–1.52 Ma), located in the mainland and the peninsula. Two additional haplogroups diverged about 1.21 Ma (95% HPD, 1.01–1.67 Ma). The first, including H2 and H5 haplotypes, diverging ca. 0.56 Ma (95% HPD, 0.01–1.63 Ma), mainly present on the mainland, and the other distributed along the peninsula and branching back 0.65 Ma (95% HPD, 0.03–1.38 Ma). The smallest haplogroup was composed of the most continental-distributed haplotypes (H4) and other mainland-restricted haplotypes (H10, H11 and H12) at 0.89 Ma (95% HPD, 0.001–2.04 Ma).

### Genetic differentiation and population structure

The STRUCTURE analysis including all populations and using $K = 2$ confirmed two geographic genetic clusters, one in the mainland and one in the peninsula (Fig 3A). Some individuals had an inconsistent genetic signature indicating genetic exchange between MAIN and PEN populations, particularly those on the central Gulf of California coast. An analysis with $K = 3$ separated the two main population groups, and the northernmost MAIN population (Sonoyta; S2 Fig). With $K = 4$ there was an additional, well-defined cluster separating the populations of the Cape Region on the southern tip of the peninsula (Balandra and El Palmar; S2 Fig).

The STRUCTURE analyses conducted only on the PEN populations revealed that the most likely number of clusters according to the Evanno et al. [63] test was $K = 3$. One cluster was found throughout the peninsula (Fig 3A, grey), another included only the monomorphic population of Santa Rosalía (yellow), and a third clustered the two Cape Region populations (Balandra and El Palmar, light purple). For the mainland, the most likely number of clusters was $K = 4$: one including widely distributed individuals (red), a second cluster with plants only from Kino (blue), a third group including thornscrub plants from Tecoripa and Zacate Blanco

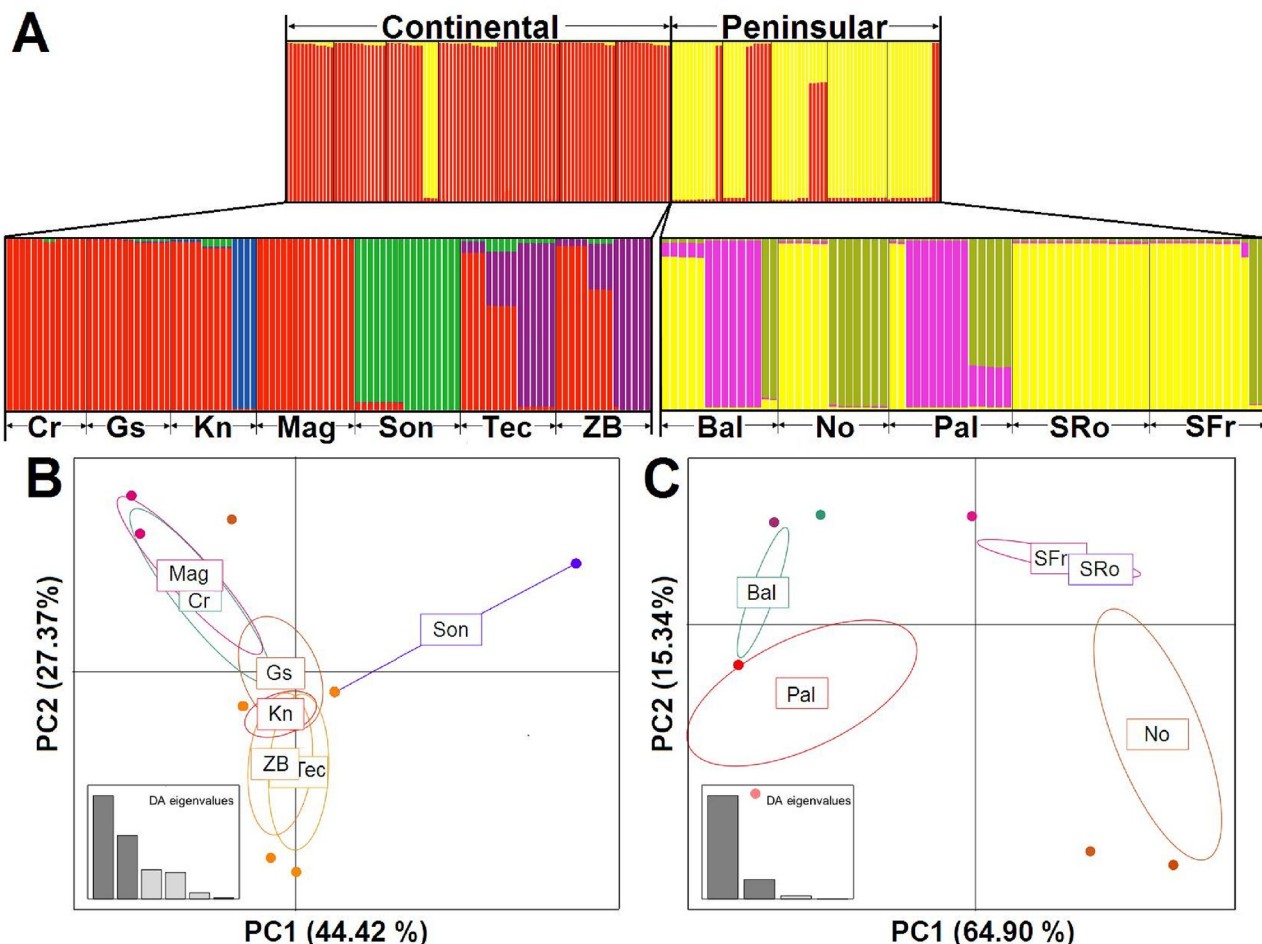

**Fig 3. STRUCTURE assignment of samples.** Assignment of the 176 samples according to geographic groups for $K = 2$ [using admixtures and correlated allele frequencies], for samples from the mainland [left, $K = 4$], and samples from the peninsula [right, $K = 3$] (A). PCA ordination of the individuals of the mainland (B), and peninsular geographic region according to their genetic structure (C) on the first two principal components [percentage variance explained in parentheses].

(purple), and a cluster dominating in the northernmost population of Sonoyta (green). These individual assignment patterns were consistent with the genealogy (Fig 3B).

The subgroups detected by the PCA in the MAIN populations differentiated the coastal populations, the inland populations (with higher values along PC2), and the northern population of Sonoyta (Fig 3B). Also, in the PEN populations three subgroups emerged: the Cape Region (Balandra and El Palmar), central (Nopolo), and northern (San Francisco and Santa Rosalía) (Fig 3C). Consistent with the results above, an AMOVA analysis showed significant genetic differences between geographic groups ($F_{CT} = 0.443^{**}$), between populations within groups ($F_{SC} = 0.367^{**}$), and within populations ($F_{ST} = 0.648^{**}$). The difference between the two groups explained 44.4% of the total variation of cpDNA while 35.2% was attributed to differences between samples within populations; and the remaining (20.5%) was attributed to differences between populations within groups. Total genetic differentiation (0.64) indicates a high genetic structure (S4 Table). On the other hand, the $G_{ST}$ value was moderate for the MAIN populations (0.143), and higher for the PEN populations (0.358) and for the whole sample (0.313).

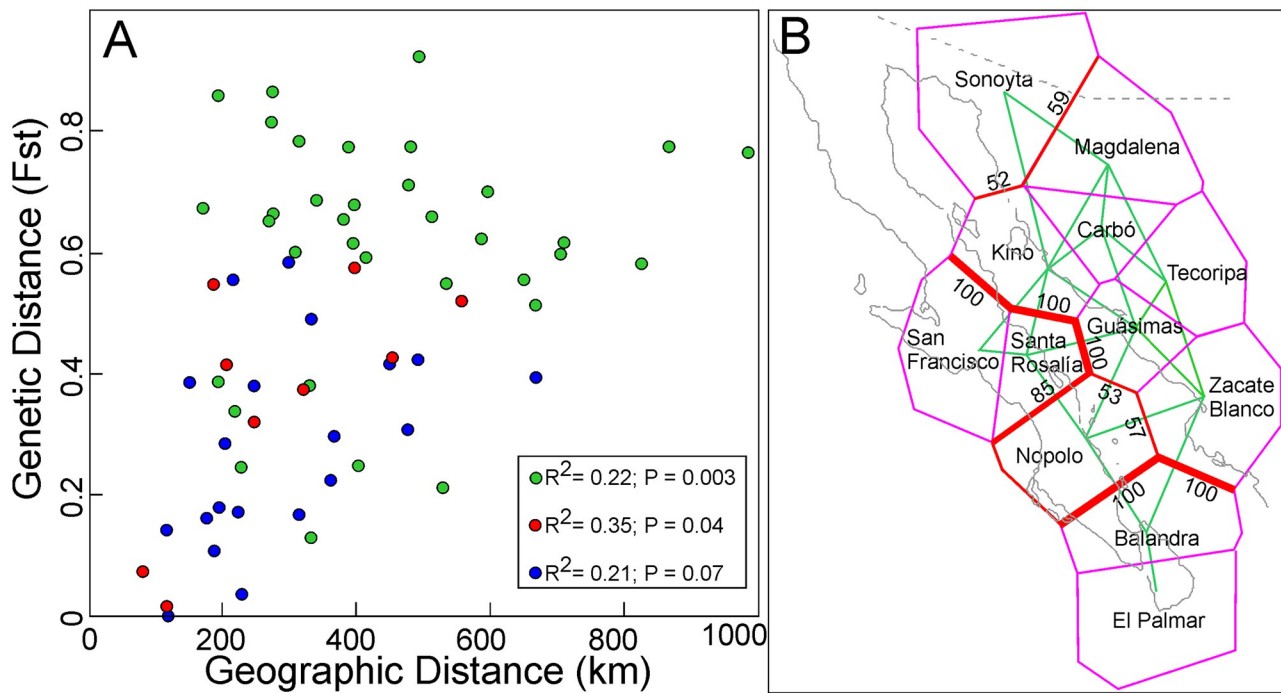

**Fig 4. Correlation between the paired geographic and genetic distances.** All populations (green), only MAIN (blue), and only PEN (red) populations of *Stenocereus thurberi* (A). Estimation of geographic barriers to gene flow among populations using the genetic and geographic distances (B). Delaunay triangulation of geographic distances (green), Voronoi tessellation showing the domain of each population (magenta). The thickness of each edge (red) and the number in the barriers is the percentage of times it was included in one of the 200 bootstrap repetitions to compute barriers. All other values in color magenta are less than 10.

The Mantel correlation between genetic and geographic distance matrices detected significant correlation for the set of populations ($R^2 = 0.22$; $P = 0.003$; $n_m = 66$) and for the PEN populations ($R^2 = 0.35$; $P = 0.04$; $n_m = 10$), but it was near the limit of statistic acceptance for the MAIN populations alone ($R^2 = 0.21$; $P = 0.07$; $n_m = 21$) (Fig 4A; S5 Table).

Using the Monmonier maximum difference algorithm on the Nei's genetic distance matrix we found that 1) mainland populations had little isolation in terms of genetic barriers (Fig 4B). The exception, with bootstrap values just above 50%, was the northernmost mainland population of Sonoyta (Fig 4B). 2) The Gulf of California is the most important barrier to gene flow. However, within the peninsula, the northern (San Francisco and Santa Rosalía) and the southern PEN populations in the Cape Region were as isolated among them as populations on the mainland and the peninsula. The northern and southern PEN populations are separated from the rest of populations by the Gulf barrier with a bootstrap support of 100%. The central PEN population of Nopolo showed a higher connectivity across the gulf than to neighboring PEN populations to the north and south (Fig 4B).

## Coalescent analysis of divergence history

Relative population migration rates ($M$) between groups were estimated using IMa for the MAIN and PEN populations, reaching convergence with high values of effective sample sizes in all parameters (ESS>1100). IMa estimated approximate $N_e$ for the MAIN group as $3.8 \times 10^5$ individuals (90% HPD = $1.6 \times 10^5$–$1.4 \times 10^6$) and $N_e$ for the PEN group as $2.7 \times 10^5$ individuals (90% HPD = $1.6 \times 10^5$–$1.6 \times 10^6$), making the mainland $N_e$ slightly larger (S3 Fig), while

ancestral $N_e$ was about 3.8 x $10^5$ individuals (90% HPD = 1.7 x $10^5$–1.1 x $10^6$). The estimated migration between the two population groups was higher than one migrant per generation (S3 Fig). The effective number of migratory events per generation from the peninsula to the mainland (PEN → MAIN), $2NM$, was 2.3 (95% HPD 0.74–7.2 migrants per group per generation), while migratory events from the mainland to the peninsula (MAIN → PEN) were estimated at 1.3 (95% HPD 0.24–5.8 migrants per group per generation). Both estimates broadly overlap, suggesting bidirectional gene flow across the Gulf of California.

## Ecological niche profiles

The estimated ecological niche models (ENM) predicted the current distribution of *S. thurberi* (Fig 5) with high certainty (AUC values >0.97 for training and testing data). Such value indicates some model overfitting, but Jackknife tests of the importance of variables to the predictive power of the model revealed that precipitation of the driest quarter, precipitation of the coldest quarter, and annual precipitation were the top three ranked variables when used in isolation. The simulation of the three periods, current, Last Glacial Maximum (LGM, 21 Ka, MIROC model), and Last Interglacial (LIG, Sangamonian stage, 120–100 Ka) times indicated extensive changes in the distribution of *S. thurberi*. The present niche modeling shows that the climatically suitable habitat for the species coincides with the current geographic distribution (Fig 5A). These analyses show a possible range shift over time, suggesting that suitable habitat for the species was reduced at the LGM.

Under the modeled climatic conditions of the LIG period, climatically suitable habitat for *S. thurberi* was present in both areas. In the mainland, a suitable patch included areas where the Kino and Guásimas populations are currently found. The second, much larger patch, included the populations of the north-central peninsula (San Francisco, Santa Rosalía and Nopolo; Fig 5C).

Stepwise multiple linear regressions of the average genetic diversities (*Hd* and π) and the main climatic parameters identified in the PCA showed that seasonality and mean annual temperature were significantly correlated with π, while Hd was only related to annual precipitation (S6 Table). When doing the analysis separately for PEN populations, annual precipitation was a crucial factor determining π and *Hd*. In the case of *Hd*, mean annual temperature also contributed with significant added variance (S6 Table). In the mainland, annual rainfall was not significant when using linear regressions, and isothermality and mean annual temperature explained a significantly large proportion of the variance in *Hd* (S6 Table). Annual precipitation is not linearly related to genetic diversity, but a non-linear inverse fit shows that precipitation is a very good predictor of π and *Hd* across populations (S2 and S6 Tables).

## Discussion

### The role of vicariance and long-distance dispersal

Different models have been proposed to explain the evolutionary processes producing the diversity and endemism of flora and fauna of the Baja California peninsula and the many islands product of the tectonic forces shaping the evolution of the Gulf [23, 31, 64]. In the case of Baja California, there is agreement that at least three major fragmentation processes associated with the creation of the Upper Gulf of California occurred since the Late Pliocene. Such processes also created the Isthmus of La Paz, which was covered by water during this same period, and the central region of Baja California, which before the Middle Pleistocene was either an island separated from the northern reaches by a shallow canal or a narrow isthmus northward [24, 30]. These processes have been ascribed as the cause for the numerous cases of endemism [27, 65, 66]. Many of them are undoubtedly the product of vicariance [8, 67].

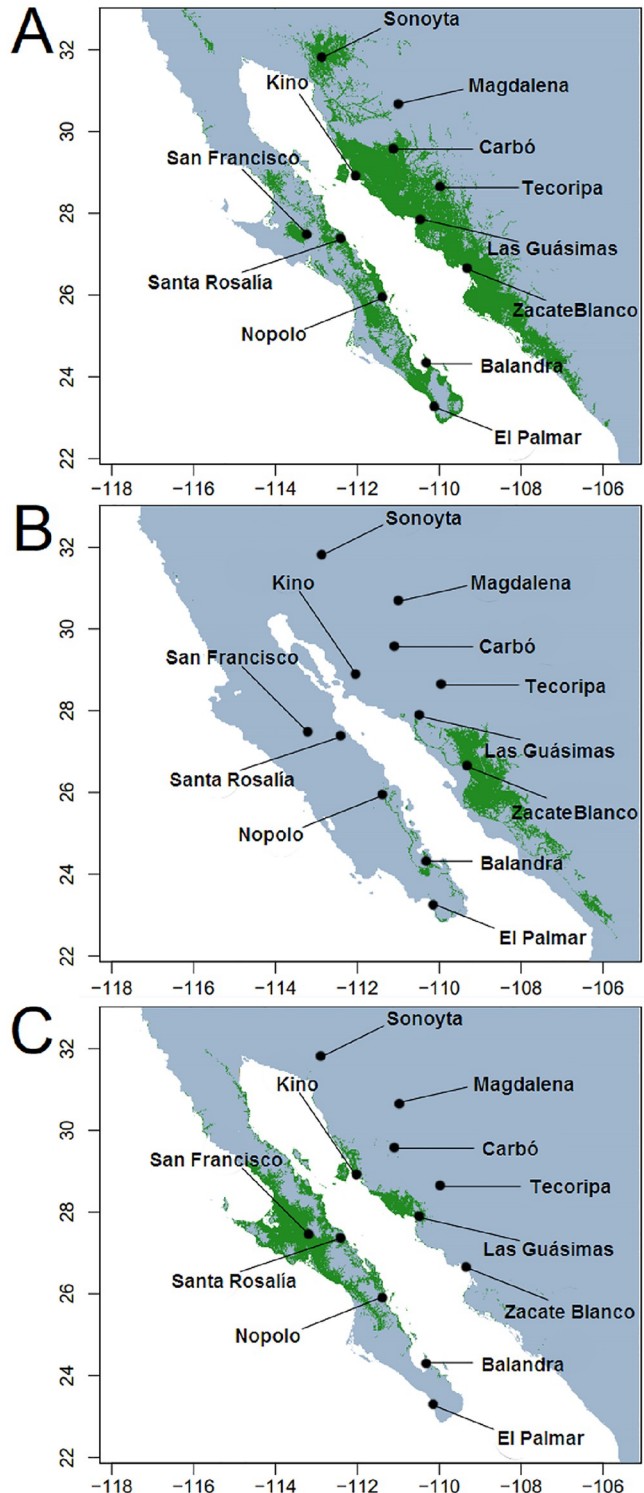

**Fig 5. Predicted environmental suitability for *Stenocereus thurberi* using ecological niche models (ENM).** ENM results are shown for: (A) the current, (B) the last glacial maximum [LGM, 21 Ka, MIROC-ESM model], and (C) the last interglacial period (LIG, 120–100 Ka) time periods.

Among these, is the remarkable case of mammilloid cacti where vicariance lead to sister species across the Gulf [68]. However, there are plant species on both coasts (and islands) like *Fouquieria columnaris* [32], *Pachycereus pringlei*, and many others that exhibit little or limited morphological differentiation, despite the geographic barrier of the Gulf of California. This similarity may indicate: 1) that genetic separation might be high but is not apparent at a morphological level (cryptic vicariance; [20]), or 2) that for some species, the Gulf of California is only a partial barrier for gene flow [31]. Any vicariance model of ancestral biota by allopatric isolation during the late Neogene would be the product of population differentiation between species on both coasts of the Gulf, including reciprocal monophily [69–71]. In these cases, the divergence would be the result of geographical barriers, but it would be difficult to exclude recent dispersal [71, 72], and gene flow.

*Stenocereus thurberi* is exposed to a variety of biotic and abiotic pressures, including seedling predation [36], drought and temperature extremes, and large disturbances from cyclonic winds and rains. Taken together, the results from *Tajima's D* test and the historical environmental fluctuations indicate apparent selective processes in Nopolo which is a potential Pleistocene refuge, and in Sonoyta, the northernmost, and probably one of the younger populations. All other populations do not show significant departures from neutrality. Spatially variable selection has been reported in other Neotropical species, the results for *S. thurberi* point in a similar direction [73]. Still, evidence of local adaptation needs to be assessed and the intensity and direction of selection quantified, requiring comparison with phenotypic traits and assessment of survival over time [74].

**Ecological niche modeling and evolution.** Niche models show that the climatically predicted habitat for the species matched the current geographic distribution of the species (Fig 5A; [26, 35]), suggesting that suitable habitat available to the species was substantially reduced during the LGM. Therefore, refugia were likely located in coastal areas of the southwestern Sonoran Desert. This location may be explained by the lower sea levels, maritime climate, and milder temperatures in southern coastal locations where the Las Guásimas, Zacate Blanco, El Palmar, Nopolo, and Balandra populations are found today (Fig 5B). Over time, increasing suitable habitat, inland and northwards, led to the present distribution range. Thus, it is likely that populations showing high genetic diversities such as Zacate Blanco and Nopolo retain the ancestral haplotypes and were Pleistocene refuges. A similar pattern was found by Sanderson et al. [75] for saguaro. Expanding studies to other populations and using information from the whole-genome throughout the species' natural range should be of interest (particularly those on islands) and reveal adaptive differentiation among populations [4, 76]. Of particular interest is the association of *Hd* and π with precipitation and temperature showing that *S. thurberi* thrives in sites with a well-defined and more predictable monsoon in the fringes of the Sonoran Desert where thornscrub and tropical dry forests develop [77, 78]. The wide environmental differentiation across the peninsula suggests local adaptation to different water regimes. The negative relationship with elevation in the peninsula reflects perhaps the effects of geodispersal from coastal populations in the mainland and the permanence of ancient populations about 125 m below the actual sea level during the last glacial maximum when low-lying plains were extensive along much closer coasts [27].

The genetic diversity described in our study is high compared to other studies using plastid markers. High levels of genetic diversity have been associated with long-lived woody plants, having high population density, long-distance dispersal of seeds and/or pollen, and extensive geographic ranges [79]. The mean genetic diversity on the mainland was slightly higher than that found in the peninsula (*Hd* = 0.813 vs. 0.713, respectively). Population differentiation was found to be remarkable in the peninsula, while the mainland populations showed less structure (*G_{st}* = 0.358 vs. 0.142, respectively), apparently because gene flow is higher on the mainland. In

the Cape Region, the peculiarly different climatic and vegetation determinants [80] and the remoteness, apparently led to restricted migration and differentiation. In the southern tip of the peninsula, back in 1892, Katharyne Brandegee recognized a new taxon, *Stenocereus thurberi ssp. littoralis* (Brandegee) N. P. Taylor [81], growing on steep coastal bluffs in a narrow area between San Jose del Cabo and Cabo San Lucas. This subspecies is much smaller, has thinner, greyish stems, pink to magenta flowers, and much smaller fruits than typical *Stenocereus thurberi*. These factors add to the idea of restricted gene flow, isolation, and differentiation as pointed out by Gibson [82] and support the notion that the Cape Region was isolated by the hypothetical flooding of the isthmus of La Paz [20]. At the same time, the differentiation of the northern populations of the peninsula back the view of an ecologically (in geography, geology, and climate) isolated area somewhere north of La Paz and south [21] of Santa Rosalía.

## The role of bats in shaping the genetic structure

The genetic structure of populations is intimately linked to the activities of their mutualistic pollinators and seed dispersers, including local and migratory movements [83]. The role of chiropterophily and chiropterochory in structuring or connecting populations of *S. thurberi* seems especially important. Nectar-feeding bats are a crucial vector for the pollination and seed dispersal of many columnar cacti [84, 85]. *Leptonycteris yerbabuenae*, the main pollinator of this species [33], and other phyllostomid bats, are efficient pollinators and seed dispersers facilitating the exploration and colonization of new areas [85, 86]. Also, several species of perching birds like orioles, grackles, and woodpeckers (but not doves, as shown by [87]) are known to disperse the seeds (Búrquez pers. obs.) but there is not yet a detailed study of frugivory and dispersal of seeds in any of the columnar cacti of NW Mexico and SW USA.

The high genetic diversity and gene flow we detected within the mainland portion of the distribution range of *S. thurberi*, reveals the absence of strong geographic barriers indicating limited genetic differentiation, and reduced isolation by distance in the mainland populations. For example, the neighboring populations of Carbó and Magdalena share haplotype 5, and Tecoripa and Zacate Blanco share haplotype 11, while the least diverse population (Sonoyta), a marginal population in the northernmost, driest portion of the range, has only haplotype (H10), suggesting ancient colonization and founder effects, geographic isolation, or ecological constraints leading to local adaptation and differentiation [88, 89]. Baja Californian populations are probably vicariant. They have been isolated from the mainland populations for about 2 million years. However, ongoing gene flow between both groups suggests that even when geographically allopatric, with the Gulf of California as a barrier to dispersal, extensive gene flow is still happening, and more than one effective migrant is exchanged between subpopulations per generation. That number of dynamically active migrants is theoretically sufficient to prevent genetic drift leading to population divergence [9].

## Genetic structure suggests gene flow across the Gulf of California

Migratory bats are known for their long-range and complex migratory routes [90]. For example, the straw-colored fruit bat, has been shown to traverse long distances across fragmented landscapes and to disperse small seeds by endozoocory across distances up to 50 km [91]. In the case of *L. yerbabuenae*, researchers have found evidence of large, migratory populations on the peninsula, suggesting that bats regularly go across the Gulf of California, potentially using the midriff islands as stepping stones [95, 96]. For example, at foraging sites, Medellín et al. [86] marked individuals of *L. yerbabuenae* with colored powder finding that they can fly up to 100 km from their roost cave to their feeding sites. This flight distance exceeds all known distances of other phyllostomids or nectarivores in the world. Examining the distribution of

genetic diversity across mainland populations of the *L. yerbabuenae* bat colonies, Ramírez [92] found two clades, but little geographic structuring, and recently Arteaga et al. [93, 94] found the same lack of pattern among peninsular colonies suggesting high levels of gene flow mediated by females. Although there are no direct observations of *L. yerbabuenae* actually using the midriff islands as stepping stones, or flying across the Gulf to reach the peninsula, there are records of this species flying from Tiburon island to the mainland in foraging bouts of about 30 km [95]. Lesser long-nosed bats fly up to 40 km to hummingbird feeders in southern Arizona [96]. Record-breaking long-distance one-night flights were documented recently by Goldshtein et al. [97]. They documented round trip flights from their roosts to their foraging grounds of up to 200 km in the Pinacate y Gran Desierto de Altar biosphere reserve. All these data suggest that *L. yerbabuenae* can fly among islands of the Gulf of California to forage and roost, or to go across the Gulf in one bout. However, if there is a paucity of data on bat flight distances, there is an even greater paucity of documented data on bat seed dispersal distances.

Migration of *L. yerbabuenae* bats between the mainland and the peninsula entail crossing the Gulf of California during the foresummer to find floral resources of columnar cacti during the flowering time early in the season when wind patterns follow a NW-SE direction [98] or later in the season at the onset of the monsoon when strong reversal of the normal surface winds occur and wind follows a roughly S-N direction in the lower Gulf (Fig 6). There is ample evidence for migratory birds taking advantage of winds to assist migration [99], and also for bats [100]. An important additional study would be to examine the hypothetical kinship among populations by haplotype presence and why it shows a remarkable similarity to the summer wind patterns, considering that the northern coastal populations are probably

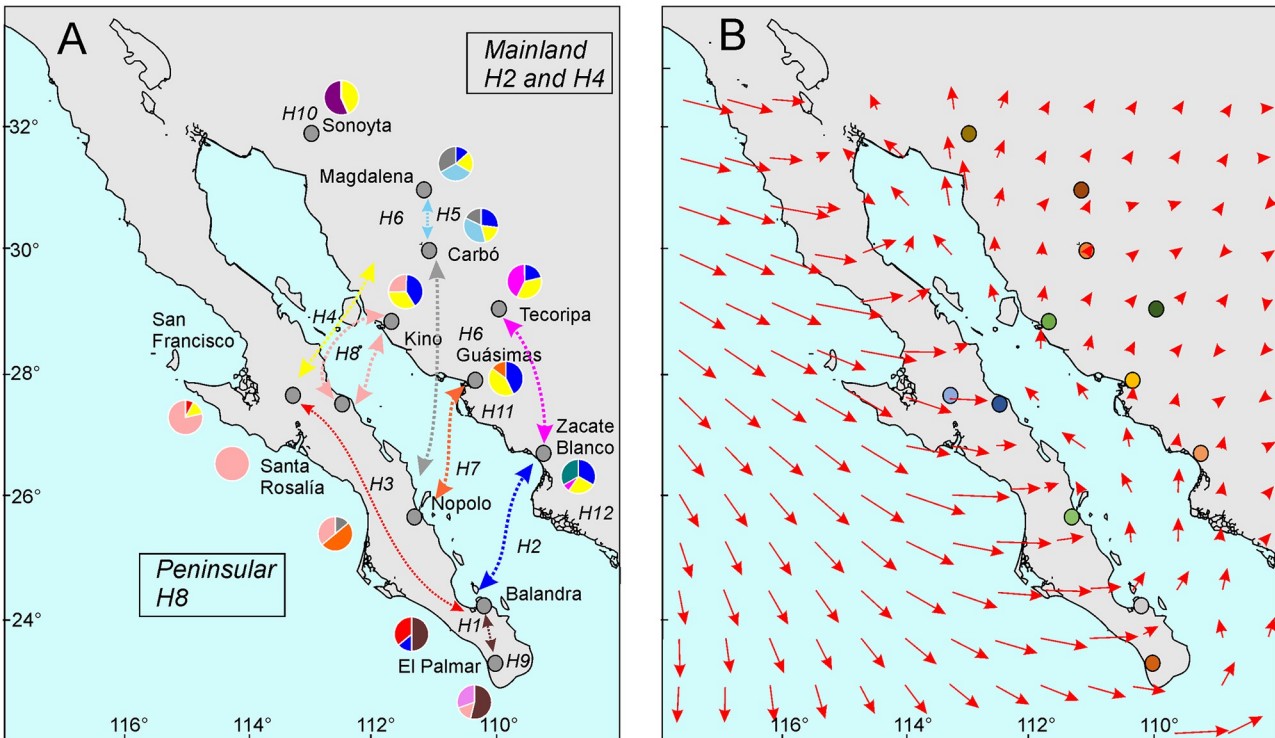

**Fig 6. Hypothetical gene flow patterns among populations of *Stenocereus thurberi* across the Gulf of California.** (A) haplotype presence, (B) surface wind patterns during summertime as modeled by Morales-Acuña et al. [98]. During early autumn, winds return to a generalized pattern like the one on the lower left side of (B).

associated with earlier bat migration across the gulf using the midriff islands as stepping stones, and then within the peninsula. The southern populations on both coasts are related by the likely direct flight of lesser long-nosed bats across the gulf. This hypothetical scenario suggest 1) an early bat migration using either the midriff islands to access floral resources, or taking advantage of the strong winds of the foresummer from the northwest to reach the tip of the peninsula, 2) a return of *L. yerbabuenae* to the mainland to continue foraging on a more diverse set of resources taking advantage of the reverse monsoon winds as noted by the late arrival of *L. yerbabuenae* to the Zacate Blanco population by Bustamante et al. [34], and 3) a return to their wintering grounds late in the season once the wind patterns revert to their autumn—winter pattern. This hypothetical scenario explains the relatedness of populations across the gulf and resolves the larger number of organ pipe cactus migrants from the peninsula to the mainland.

The divergence between the peninsular ancestral population occurred approximately 1.56 million years and Pleistocene refugia were present in both population groups. In *S. thurberi*, two phylogeographic histories occur, one in the mainland where extensive gene flow generates greater homogeneity between populations in the form of a network, and another in the peninsula where isolation by distance favors greater genetic divergence. Still, evidence of gene flow within mainland populations, and between the mainland and the peninsula is compelling.

## Supporting information

**S1 Table. Mainland and peninsular populations used in this study.** Name, geographical location, elevation, and vegetation type of *Stenocereus thurberi* populations used in this study. The peninsular group are populations in Baja California Sur (México) and the mainland group are populations in Sonora (México).
(DOCX)

**S2 Table. Coefficients of determination for each variable in the first three principal components for the niche model of *Stenocereus thurberi*.** The PC1 was most affected by precipitation bioclimatic variables, while seasonal precipitation and annual temperature had the most influence on the PC2. For PC3, the maximum temperature of the warmest month, seasonality, and isothermality contributed with significant variance.
(DOCX)

**S3 Table. Haplotype distribution among mainland and peninsular populations of *Stenocereus thurberi*.** Pink shade, absent from that group. Blue shade, most common within a group. Green shading, unique haplotypes.
(DOCX)

**S4 Table. Summary of analysis of molecular variance in populations of *Stenocereus thurberi*.** The AMOVA was performed with three molecular markers of chloroplast.
(DOCX)

**S5 Table. Matrix of geographic and genetic distances among populations of *Stenocereus thurberi*.** Below the diagonal, distances in km, above the diagonal, genetic differentiation [pairwise $F_{st}$].
(DOCX)

**S6 Table. Stepwise multiple regression equations relating π and *Hd* with the worldclim variables.** The multiple regressions were used to find the most influential variables on determining the ecological niche model for *Stenocereus thurberi*. Subscript all = all populations,

n = 12, P = only peninsular, N = 5, C = only mainland, n = 7.
(DOCX)

**S1 Fig. Relationship between haplotype diversity [Hd] and elevation [A], and rainfall [B].**
Red = mainland populations, Blue = peninsular populations of *Stenocereus thurberi*.
(TIF)

**S2 Fig. Gene structure of *Stenocereus thurberi* derived from STRUCTURE analyses.** The
individual proportion of membership shown for three, four and five clusters [from top to bottom]. The codes of the populations are the same as those in S1 Table.
(TIF)

**S3 Fig. Posterior probabilities density for demographic parameters between mainland and
peninsular population groups of *Stenocereus thurberi*.** Demographic estimates were calculated by the isolation with migration model of Ossowski et al. [57] mutation rate [$7 \times 10^{-9}$
mutations per site per year], and 20 y generation time. A) Estimates of the effective population
size for ancestral, mainland, and peninsular populations. B) Population migration rates, mainland → peninsula [blue] and peninsula → mainland [red]. HPD = Highest Posterior Density.
(TIF)

**S1 Methods. Methods for the full PCR procedure.**
(DOCX)

**S1 Striking image.**
(GIF)

**S2 Striking image.**
(GIF)

## Acknowledgments

The authors thank Dr. Valeria Souza for her assistance in the field work, and Drs. Erika
Aguirre-Planter, Laura Espinosa-Asuar, Santiago Ramírez-Barahona and Josué Barrera-Redondo (all at IE-UNAM) for their valuable help in the laboratory analysis. We also thank
Josué Barrera-Redondo, Gustavo Giles and Jonás Aguirre-Liguori for their help with the bioinformatics analyses. Dario Copetti, Exequiel Ezcurra, and one anonymous reviewer made valuable suggestions to the manuscript.

## Author Contributions

**Conceptualization:** Sebastián Arenas, Alberto Búrquez, Enrique Scheinvar, Luis E. Eguiarte.

**Data curation:** Sebastián Arenas.

**Formal analysis:** Sebastián Arenas, Alberto Búrquez, Enrique Scheinvar.

**Funding acquisition:** Alberto Búrquez.

**Investigation:** Sebastián Arenas, Alberto Búrquez.

**Methodology:** Sebastián Arenas, Alberto Búrquez, Enriquena Bustamante, Luis E. Eguiarte.

**Project administration:** Enriquena Bustamante.

**Resources:** Sebastián Arenas.

**Software:** Sebastián Arenas, Enrique Scheinvar.

**Supervision:** Alberto Búrquez, Enrique Scheinvar, Luis E. Eguiarte.

**Validation:** Sebastián Arenas, Alberto Búrquez, Luis E. Eguiarte.

**Visualization:** Sebastián Arenas.

**Writing – original draft:** Sebastián Arenas, Alberto Búrquez.

**Writing – review & editing:** Sebastián Arenas, Alberto Búrquez, Enriquena Bustamante, Enrique Scheinvar, Luis E. Eguiarte.

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
