## [Decision Letter · Decision Letter 0]

23 Mar 2023

PONE-D-23-05594Are 150 km of open sea enough? Gene flow and population differentiation in a bat-pollinated columnar cactusPLOS ONE

Dear Dr. Búrquez,

Thank you for submitting your manuscript to PLOS ONE. After careful consideration, we feel that it has merit but does not fully meet PLOS ONE’s publication criteria as it currently stands. Therefore, we invite you to submit a revised version of the manuscript that addresses the points raised during the review process.

Both reviewers had a favorable response to your manuscript and noted only minor revisions were necessary.  Some rewriting and reworking of the Intro and Discussion is needed to improve clarity, flow, and to add missing details, but shouldn't be too time consuming to accomplish. 

We look forward to receiving your revised manuscript.

Kind regards,

Dr. Janice L. Bossart

Academic Editor

PLOS ONE

Journal Requirements:

"No"

4. We note that Figure 1, 4, 5 and 6 in your submission contain [map/satellite] images which may be copyrighted. All PLOS content is published under the Creative Commons Attribution License (CC BY 4.0), which means that the manuscript, images, and Supporting Information files will be freely available online, and any third party is permitted to access, download, copy, distribute, and use these materials in any way, even commercially, with proper attribution. For these reasons, we cannot publish previously copyrighted maps or satellite images created using proprietary data, such as Google software (Google Maps, Street View, and Earth). For more information, see our copyright guidelines: http://journals.plos.org/plosone/s/licenses-and-copyright.

1. You may seek permission from the original copyright holder of Figure 1, 4, 5 and 6 to publish the content specifically under the CC BY 4.0 license.  

Additional Editor Comments (if provided):

Reviewers' comments:

Reviewer's Responses to Questions

**Comments to the Author**

1. Is the manuscript technically sound, and do the data support the conclusions?

Reviewer #1: Yes

Reviewer #2: Yes

2. Has the statistical analysis been performed appropriately and rigorously? 

Reviewer #1: Yes

Reviewer #2: Yes

3. Have the authors made all data underlying the findings in their manuscript fully available?

Reviewer #1: Yes

Reviewer #2: Yes

4. Is the manuscript presented in an intelligible fashion and written in standard English?

Reviewer #1: Yes

Reviewer #2: Yes

5. Review Comments to the Author

Reviewer #1: An excellent, well done and interesting study. I suggest streamlining the introduction and background and explanding the discussion. In the discussion, the significance of several results seems to not be covered in enough detail. The discussion would be improved by using subheadings that are in the same order as the presentation of results.

Reviewer #2: The Gulf of California is an extremely interesting oceanic barrier that dramatically divides the Sonoran Desert in two: the mainland deserts and the Baja California Peninsula. Both subdivisions are biologically very distinct, possibly due the 6-5 million years of isolation imposed by the geologic rift, and by the complex interplay of spatial isolation and gene flow across the Gulf. This uses the distribution and genetic traits of the organ-pipe cactus (Stenocereus thurberi) to infer the evolution of the species during the late Pleistocene, the role of the species’ pollinators and seed dispersers in gene flow across the Gulf, and to understand the current status of the species’ overall genetic diversity and genetic differentiation into local populations.

The driving hypothesis of the paper is extremely interest and scientifically relevant, the methods are robust and carefully developed, and the results add a very important piece of knowledge to our current understanding on the evolution of the unique biota of the Baja California Peninsula.

Minor edits and corrections are attached in the edited comments to the pdf of the original manuscript.

6. PLOS authors have the option to publish the peer review history of their article (what does this mean?). If published, this will include your full peer review and any attached files.

Reviewer #1: No

Reviewer #2: **Yes: **Exequiel Ezcurra

---

## [Author Response · Author response to Decision Letter 0]

3 May 2023

Response to reviewers

We appreciate the comments and queries made by the reviewers. These significantly improved the manuscript. We attended to all suggestions. Two citations were removed and several paragraphs deemed unnecessary were deleted. As a consequence, the reference section was updated. We added text to clarify some sections as requested by the reviewers. In total, the reduction of unnecessary text and additions asked by reviewers left the manuscript almost the same size. All figures are original and do not need copyright permission for publication. Figure 6 is redrawn from public data and the authors are acknowledged. 

We confirm this work has not been published elsewhere, nor is it currently under consideration for publication, nor do we have conflicts of interest to disclose.

Abstract:

Corrected species name in italics.

Introduction:

Line 46, We described explicitly the query by the reviewer on founder effects

Following the suggested abridging and focusing of the introduction led us to reduce the content by deleting the following paragraphs:

Lines 49-52. The repetitive paragraph was deleted, including citation number 5.

Lines 60-64. Unnecessary information. The paragraph and citation 13 were deleted. 

In line 64 we deleted "In" 

line 88 deleted citation 23 (Karig), deleted ", and reduced gene flow"

line 115. added "]."

Line 114: Myrtillocactus cochal authorities are correct as originally stated in the ms. The basionym was described by C.R. Orcutt in 1889 and later assigned to Myrtillocactus by Britton & Rose. https://tropicos.org/name/5100588

Lines 115-116. Deleted "Their presence is likely the result of vicariant events."

Lines 119-123. Rearrangement of the sentence. Deleted as suggested "that has a highly specialized pollinator mutualism by moths [32],"

Line 123. Made clear that Pachycereus pringlei does not have an extensive distribution on the mainland.

Line 128. Added suggested comments on the narrow endemics at the tip of the peninsula (from line 114 onward).

Material and Methods:

Line 138. Added a brief description of the distribution range.

Line 167. Answered to the reviewer's question "Did you find you also had to manually adjust the alignment?" is yes. Added at the beginning of paragraph "After removing some nucleotides in the initial and final portion of the sequences (~15 nucleotides), multiple..."

Line 222, 223, 239. changed values to scientific notation

Line 248 Query by reviewer "What spatial relationship did the occurrences have? Was it necessary to do any thinning or adjustment to match the climate scale?" Yes. We tried to attain a homogeneous cover of the distribution range and bioclimatic envelope.

Results:

Line 398. Query by reviewer "This seems like perhaps the model was overfit somewhat. Maybe because .70 is a pretty high cutoff for correlation? The use of 9 climate variables is a lot of predictors. A pretty complex model." Added a note of caution highlighting the value of the three top bioclimatic variables.

Line 411. As suggested by the reviewer, the explanation of refuges was moved to the discussion.

Line 428. Reduced and moved to discussion

Discussion:

Line 434. Added subheadings to the discussion: 1) The role of vicariance and long-distance dispersal, 2) Ecological niche modeling and evolution, 3) The role of bats in shaping the genetic structure, and 4) Genetic structure suggests gene flow across the Gulf of California

Line 472. We rearranged the paragraph and included the niche modeling discussion suggested by reviewers in line 411.

Line 515. The query by the reviewer is relevant. "I guess it doesn't seem necessary to state the direction of dispersal? It seems highly unlikely that thurberi would have originated on the peninsula. But I could be wrong." We added a sentence explaining the possible evolutionary forces acting upon these populations.

Line 517. Made clear that Baja Californian populations are likely vicariant after the extensive separation of the peninsula.

Line 547. Made clear the lack of data on bat flight and seed dispersal.

Line 559. Agree. We might attempt it for the cardón sahueso (Pachycereus pringlei), widespread species of the islands.

Line 569. As suggested by the reviewer, we added a short paragraph summarizing the main findings

---

## [Editor Report · Decision Letter 1]

15 May 2023

PONE-D-23-05594R1Are 150 km of open sea enough? Gene flow and population differentiation in a bat-pollinated columnar cactusPLOS ONE

Dear Dr. Búrquez,

Thank you for submitting your manuscript to PLOS ONE. After careful consideration, we feel that it has merit but does not fully meet PLOS ONE’s publication criteria as it currently stands. Therefore, we invite you to submit a revised version of the manuscript that addresses the points raised during the review process. Both reviewers had very favorable responses to your manuscript and noted it was interesting and an important study.  Congratulations!  Reviewer 1 has identified multiple areas where text needs to be modified/rewritten to improve clarity and flow, and to ensure key results are discussed.  Thoughtful rewriting will take a bit of time but shouldn't be too onerous.  

We look forward to receiving your revised manuscript.

Kind regards,

Dr. Janice L. Bossart

Academic Editor

PLOS ONE
---

## [Author Response · Author response to Decision Letter 1]

18 May 2023

We attended all of the queries and included all corrections made by reviewers. In particular, in the response letter we detail how we included in the manuscript Reviewer 1 research questions and how we improved, with the aid of a copy editor, these areas where reviewer 1 found our text needed to be modified or rewritten to improve it. Reviewer's comments were valuable in ensuring that our key results were discussed. Thanks to their effort, we ended with a more concise and clear article.

---

## [Editor Report · Decision Letter 2]

29 May 2023

Are 150 km of open sea enough? Gene flow and population differentiation in a bat-pollinated columnar cactus

PONE-D-23-05594R2

Dear Dr. Búrquez,

We’re pleased to inform you that your manuscript has been judged scientifically suitable for publication and will be formally accepted for publication once it meets all outstanding technical requirements.  Congratulations!  It's a very interesting study.  The modifications you've made to increase flow and clarity are much appreciated.  Please note that PLOS ONE has specific requirements for Supporting Information, e.g. a list of captions at the end of the manuscript, and strongly recommends that each caption has a title (https://journals.plos.org/plosone/s/supporting-information).  I know I personally prefer titles when I'm reading an article and accessing any supporting files.

Kind regards,

Dr. Janice L. Bossart

Academic Editor

PLOS ONE
---

## [Editor Report · Acceptance letter]

19 Jun 2023

PONE-D-23-05594R2 

Are 150 km of open sea enough? Gene flow and population differentiation in a bat-pollinated columnar cactus 

Dear Dr. Búrquez:

I'm pleased to inform you that your manuscript has been deemed suitable for publication in PLOS ONE. Congratulations! Your manuscript is now with our production department. 

Kind regards, 

on behalf of

Dr. Janice L. Bossart 

Academic Editor

PLOS ONE